# High-Precision Flow Field Simulation of Aerostatic Bearings Based on the Interior Penalty Discontinuous Galerkin Method

Weijie Hou [1,2], Qiushi Ding [3,4], Yongbo Hao [5], Jingshuo Cao [3,4], Shixi Hao [3,4], Kai Feng [1] and Ming Zhao [3,4,*]

1   College of Mechanical and Vehicle Engineering, Hunan University, Changsha 410082, China
2   Tianjin Institute of Aerospace Mechanical and Electrical Equipment, Tianjin 300301, China
3   Department of Mechanics, School of Mechanical Engineering, Tianjin University, Tianjin 300350, China
4   Tianjin Key Laboratory of Modern Engineering Mechanics, Tianjin University, Tianjin 300350, China
5   Beijing Institute of Control Engineering, Beijing 100190, China
*   Correspondence: ming.zhao@tju.edu.cn

**Abstract:** All the static performance, dynamic characteristics, and stability are strongly associated with the flow field inside the aerostatic bearings. Therefore, a high-precision numerical method is beneficial for the detailed description of the bearing flow field. To this end, a modified interior penalty discontinuous Galerkin method was introduced here. Actually, a lift operator was included to eliminate the so-called homogeneity tensor connecting the viscous term and variable gradient, which could improve the numerical feasibility. The accuracy of the above numerical method has been comprehensively validated through viscous cases, including Couette flow and shear-driven cavity flow. Then, the flow fields of three aerostatic bearings were simulated with different orifice geometries. As a result, the Mach number distributions and static pressure could be estimated together with the integration of the pressure acting upon the thrust surface. The acceleration within the orifice and air film could be detected, and the influence of the orifice geometry has been systematically discussed.

**Keywords:** aerostatic bearing; discontinuous Galerkin method; computational fluid dynamics

## 1. Introduction

Aerostatic bearings have been widely adopted in various industries, such as electronics and semiconductors [1], metrology and ultra-precision machine tools [2], turbomachinery [3], the medical industry [4], etc. They always encounter extremely high working requirements, including high precision, high stability, and high speed. Therefore, it remains challenging to develop high-performance aerostatic bearings [5].

All the static performance, dynamic characteristics, and stability are strongly associated with the flow field inside the aerostatic bearing. Therefore, it is extremely important to simulate the details of the flow field accurately, including shock wave/boundary layer interaction, rarefaction wave, shear layer, etc. [6].

Traditionally, many flow models have been proposed to analyze the performance of aerostatic bearings, which always introduce assumptions to simplify the governing equations [7]. Therefore, the intrinsic driving force of the oscillation could not be comprehensively depicted. Along with the development of computational fluid dynamics (CFD), numerical simulations have been conducted to analyze the flow structures together with the overall performances of the bearings [8]. Turbulence models, including Reynolds-averaged Navier–Stokes (RANS) models [9] and the large eddy simulation (LES) method [7], have been systematically implemented. With the help of the CFD method, the details of flow status inside the air film, such as the pressure contour, the velocity vectors, and streamlines, could be observed. For example, three-dimensional, turbulent, Navier–Stokes simulations for compressible air flow were presented in [6], where the k-ε turbulence model was adopted. The computational flow visualization in the inlet region showed the coalescing of compression waves into shock waves, the reflection shocks, and a region of

shock/boundary layer interaction. Meanwhile, the LES method was employed to calculate the transient flow field in the bearing clearance in [7] numerically. Repeated pressure depression (in space) and fluctuation (in time) could be observed in the bearing clearance when vortex shedding occurred. Both steady and transient flow calculations were carried out in [8]. On this basis, radial basis function (RBF) models were established, and design optimization was performed. It was found that shock-wave-induced flow separation and vortices in the air film made the local pressure and the total distribution dramatically fluctuate over time.

To overcome the difficulty of describing complex geometrical configurations and fluid-structure interactions (FSI), Chen et al. proposed a dynamic mesh technique (DMT) based on CFD software to study the dynamic characteristics of multi-restrictor aerostatic bearings with shallow recesses [10]. Then, the DMT method was also applied to calculate the dynamic performance of the aerostatic thrust bearings with orifice restrictors, multiple restrictors, and porous restrictors, respectively [8].

Although the CFD method has played an important role in the flow field simulation and mechanism analysis of aerostatic bearings, most of the research is conducted within the framework of the finite volume method (FVM) and commercial software. However, along with the abrupt increase of Reynold's numbers, the requirement of delicate flow structure resolution emerges. Numerical methods with high precision could resolve more flow field details compared to the finite volume method, which has become an important tendency of CFD in the last two decades [11]. There are several candidates for high-precision calculation, among which the discontinuous Galerkin methods (DGM) have become popular for the solution of systems of conservation laws [12]. Apart from the well-known high-order characteristics, it also combines the ease of finite element approximations in handling complex geometry and adaptation with the shock-capturing abilities of the finite volume method [13]. Therefore, the DGM has many attractive features: (1) It has useful mathematical properties in terms of conservation and convergence. (2) It can be extended to a higher order (>2nd order). (3) It is well suited for complex geometries as it can be applied to unstructured grids. (4) It is highly parallelizable due to its compactness.

Recently, due to the development of BR1/BR2 [14], interior penalty (IP) [15], and reconstructed DG (RDG) methods [13], the calculation of viscous terms could be implemented within the DGM framework. Among the above methods, the IP method has been widely adopted due to its compactness and ease of implementation, which refers to the IPDG method here. This method could easily be reinterpreted as the combination of a large number of Galerkin finite element problems, which are coupled by internal boundary conditions across the interfaces [15]. For example, the application of the IPDG method to implicit LES of free and wall-bounded equilibrium turbulence has been conducted in [16]. In detail, the method has been applied to the simulation of decaying homogeneous isotropic turbulence at a very high Reynolds number and the turbulent channel flow simulation. Therefore, it is possible to deliver a high-precision simulation of the bearing system based on the IPDG method.

However, in the traditional IPDG method, there is a fourth-order homogeneity tensor connecting the viscous terms and variable gradients, which is difficult to calculate and program. In the present study, the DGM with a modified IP method was introduced to discretize the convection and diffusion terms of the Navier–Stokes equations of test cases and aerostatic bearing flow field. The accuracy has been comprehensively validated through viscous incompressible and compressible cases. To our knowledge, it is the first attempt to conduct a high-precision simulation with DGM in the field of aerostatic bearing flow field analysis. In practice, the flow fields of three aerostatic bearings have been simulated with different orifice geometries. It turns out that the orifice without contraction has the best bearing capacity, while geometrical contraction might lead to an early transition to supersonic flow within the orifice.

The remainder of the paper is arranged as follows: The numerical framework and validation of the IPDG method are presented in Section 2. The bearing case description and

mesh information are introduced in Section 3. Numerical results and analysis are reported in Section 4. Concluding remarks are given in Section 5.

## 2. Numerical Framework and Validation

As mentioned above, the IPDG method was adopted for the discretization of the compressible Navier–Stokes equations (shown in Equation (1)). We assume that the computational domain could be subdivided into shape-regular meshes. The averages and jumps at the interior edges of the mesh are denoted as $\{U\} = (U^+ + U^-)/2$ and $[U] = (U^+ \otimes n^+ + U^- \otimes n^-)$.

$$\frac{\partial U}{\partial t} + \nabla \cdot F^c(U) - \frac{1}{\text{Re}} \nabla \cdot F^v(U, \nabla U) = 0 \tag{1}$$

The vector $U$ in Equation (1) represents the conserved variables. $F^c$ and $F^v$ indicate the convection and viscous fluxes. Then, let $\varphi$ be smooth vector functions inside each mesh element. The inner product of Equation (1) with a smooth test function $\varphi$ and integrating by parts gives the following:

$$\begin{aligned} & \int\limits_{\Omega_k} \frac{\partial U_h}{\partial t} \cdot \varphi_h d\Omega + \int\limits_{\partial \Omega_k} \left( F^c(U_h) \cdot \vec{n} - F^v(U_h, \nabla U_h) \cdot \vec{n} \right) \cdot \varphi_h d\Gamma \\ & - \int\limits_{\Omega_k} \nabla \varphi_h : (F^c(U_h) - F^v(U_h, \nabla U_h)) d\Omega = 0 \end{aligned} \tag{2}$$

Here, subscript $h$ means that the equations have been rewritten into the discrete form of finite element space. Due to the integration of parts with the test function and high-order features of variable distribution, the DG method has high-precision characteristics, which are beneficial for the resolution of delicate flow structures.

Since the viscous terms are one order higher than the convection terms, an auxiliary variable $\Theta = F^v(U, \nabla U) = G_{ij}(U) : \nabla U$ is introduced here, where $G_{ij}(U)$ is a fourth-order tensor (i.e., the so-called homogeneity tensor) representing the relationship between the viscous terms and variable gradients. Then, integration by parts was conducted for the auxiliary variable.

$$\begin{aligned} & \int\limits_{\Omega_k} \Theta : \nabla \varphi_h d\Omega \\ & = \int\limits_{\Omega_k} (G_{ij} : \nabla U) : \nabla \varphi_h d\Omega \\ & = \int\limits_{\Omega_k} \nabla U : \left( G_{ij}^T : \nabla \varphi_h \right) d\Omega \\ & := \int\limits_{\partial \Omega_k} \{U\} \otimes \vec{n} : \left( G_{ij}^T : \nabla \varphi_h \right) d\Gamma - \int\limits_{\Omega_k} U \cdot \nabla \cdot \left( G_{ij}^T : \nabla \varphi_h \right) d\Omega \\ & := \int\limits_{\Omega_k} \Theta : \nabla \varphi_h d\Omega + \int\limits_{\partial \Omega_k} G_{ij} : (\{U\} - U) \otimes \vec{n} : \nabla \varphi_h d\Gamma \end{aligned} \tag{3}$$

Actually, integration by parts has been implemented twice so that the viscous term in Equation (2) could be directly updated. Different from the BR1 and BR2 methods, the governing equation of the auxiliary variable is not indispensable within the IPDG framework, which has ease of implementation. Eventually, according to our previous study [17], a lift operator has been introduced here $L = G_{ij} : (\{U\} - U) \otimes \vec{n} = F^v \left( \{U\}, -\frac{1}{2}[U] \otimes \vec{n} \right)$, analogous to the viscous term. Therefore, the fourth-order tensor $G_{ij}(U)$ could be eliminated. Therefore, the N-S equations could be expressed in the following form:

$$\int\limits_{\Omega_k} \frac{\partial U_h}{\partial t} \cdot \varphi_h d\Omega + \int\limits_{\partial \Omega_k} \left( F^c \cdot \vec{n} - F^v \cdot \vec{n} \right) \cdot \varphi_h d\Gamma - \int\limits_{\Omega_k} \nabla \varphi_h : (F^c - F^v) d\Omega + \int\limits_{\partial \Omega_k} L : \nabla \varphi_h d\Gamma = 0 \tag{4}$$

The Interior penalty method is utilized for the numerical flux of the viscous term, where $C_{ip}$ is a constant and $h_f$ is the characteristic length of the element.

$$F^v\left(U_h^-, U_h^+\right) \cdot \vec{n} = \frac{1}{2}\left(F^v\left(U_h^-\right) + F^v\left(U_h^+\right)\right) \cdot \vec{n} - C_{ip}\frac{p^2}{h_f}F^v\left(\frac{1}{2}\left(U_h^- + U_h^+\right), \frac{1}{2}\left(U_h^+ - U_h^-\right) \cdot \vec{n}\right) \cdot \vec{n} \tag{5}$$

Meanwhile, numerical fluxes of convection terms are also introduced at the edges of the elements with the help of the Lax-F scheme. The accuracy of the above numerical method has been comprehensively validated through viscous cases, including Couette flow and shear-driven cavity flow.

### 3. Couette Flow

Couette flow is a steady viscous flow with a static lower boundary and a moving upper boundary $U$ within the laminar regimes. The exact solution to the problem is

$$\begin{cases} u = \frac{y}{H}U \qquad v = 0 \\ T = T_0 + \frac{y}{H}(T_1 - T_0) + \frac{\mu U^2}{2\kappa}\frac{y}{H}\left(1 - \frac{y}{H}\right) \\ P = const \quad \rho = \frac{p}{RT} \end{cases} \tag{6}$$

where $H = 2$ is the distance between the upper and lower boundaries. The non-dimensional velocity of the upper surface is $U = 1$, while the non-dimensional temperature of the upper and lower surfaces is $T_0 = 0.8$ and $T_1 = 0.85$, respectively. Moreover, $\mu$ represents the viscosity. As shown in Figure 1, the H-topology grid system has been constructed and the information on the grid resolution is listed in Table 1. As a result, the distribution of velocity is also demonstrated in Figure 1.

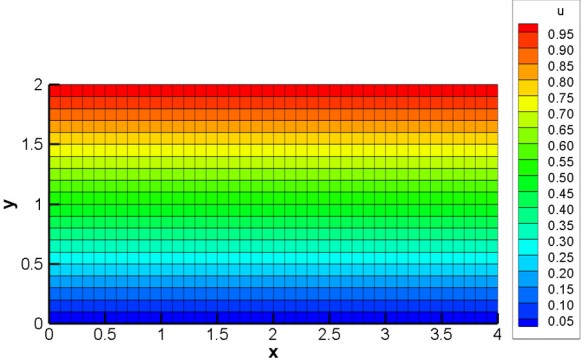

**Figure 1.** Sketches of the grid system and velocity distribution of IPDG (1).

**Table 1.** Error and numerical order of Couette flow.

| Element Number | IPDG (1) | | IPDG (2) | | IPDG (3) | | IPDG (4) | |
|---|---|---|---|---|---|---|---|---|
| | Error | Order | Error | Order | Error | Order | Error | Order |
| 50 (10 × 5) | 0.0096 | NA | $9.5 \times 10^{-5}$ | NA | $8.3 \times 10^{-7}$ | NA | $4.2 \times 10^{-9}$ | NA |
| 200 (20 × 10) | 0.0055 | 0.81 | $2.4 \times 10^{-5}$ | 2.00 | $1.1 \times 10^{-7}$ | 2.88 | $2.6 \times 10^{-10}$ | 4.00 |
| 800 (40 × 20) | 0.0029 | 0.92 | $6.2 \times 10^{-6}$ | 1.94 | $1.5 \times 10^{-8}$ | 2.95 | $1.5 \times 10^{-11}$ | 4.09 |

The accuracy of the modified IPDG method is validated with different mesh resolutions, as indicated in Table 1. The number in the brackets of the IPDG method indicates the theoretical accuracy. It could be observed that the numerical accuracy of our simulation is consistent with the ideal one. Along with the increase in numerical order, the error monotonously declines.

## 4. Shear-Driven Cavity Flow

The laminar incompressible flow in a square cavity whose top wall moves with a uniform velocity in its own plane has served over and over again as a model problem for testing and evaluating numerical techniques.

A computational mesh with 1600 (40 × 40) elements was constructed and the Reynolds number is 1000. The velocity profiles at x = 0.5 and y = 0.5 are illustrated in Figure 2. The reference profiles come from the numerical results of Ghia et al. [18]. The finite difference method was adopted, and 16,641 elements were included, which could be regarded as dense enough to obtain accurate numerical results. In general, the results of the second-order IPDG method (p = 1) deviate from the reference, the discrepancy of which is more obvious near the cavity boundaries. In contrast, the results of the third (p = 2)- and fourth (p = 3)-order are more accurate. In essence, the numerical dissipation of the second-order DG method could account for the degradation of velocity gradients near cavity boundaries.

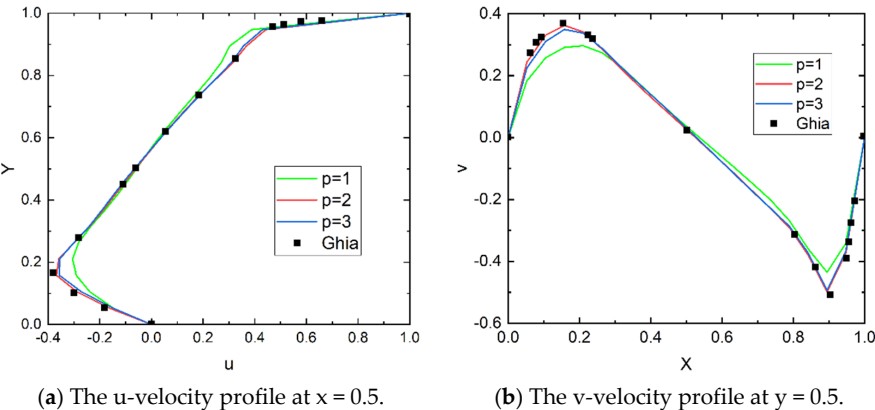

(**a**) The u-velocity profile at x = 0.5.  (**b**) The v-velocity profile at y = 0.5.

**Figure 2.** Velocity profiles of the shear−driven cavity flow field.

Based on the above simulations, the accuracy of the present high-precision IPDG method could be validated, the advantages of which could be quantitatively observed. Actually, smaller errors could be obtained with coarse mesh resolution. It will be more remarkable when industrial problems are simulated with a higher Reynolds number.

## 5. Supersonic NACA0012 Airfoil

To discuss the accuracy of shock capture, the third-order IPDG method was adopted to simulate the flow field of the NACA0012 airfoil under the conditions of $Ma_\infty = 2$, $Re = 500$, and $\alpha = 10°$. The distributions of the Mach number and pressure coefficient are demonstrated in Figure 3. The resolution of the shock wave is acceptable and the coincidence of numerical and reference pressure coefficients [19] could validate the accuracy of the modified IPDG method under compressible circumstances. Meanwhile, the influence of viscosity could be observed from the velocity gradient within the boundary layer.

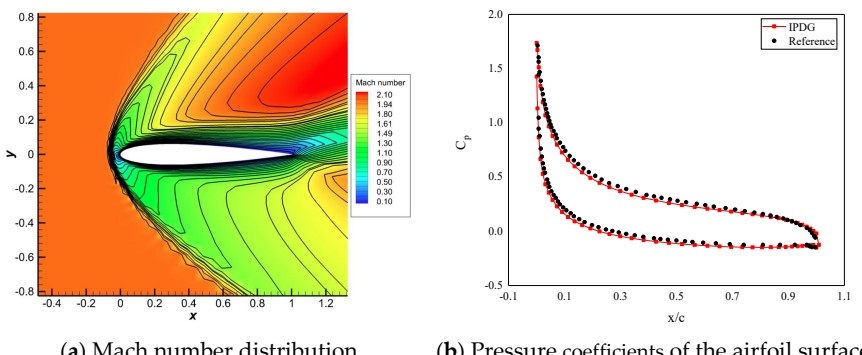

(**a**) Mach number distribution  (**b**) Pressure coefficients of the airfoil surface

**Figure 3.** Mach number and static pressure distributions.

## 6. Bearing Case Description and Mesh Information

The geometries of aerostatic bearings are illustrated in Figure 4 and Table 2. Actually, there are three different selections in the present study. The diameter of the inlet is 1 mm and the clearance height h = 0.035 mm for all the cases. The difference mainly lies in the geometrical parameters of the orifice. In detail, the diameters of the upper and lower surfaces of the orifice are listed in Table 2. Therefore, the influence of orifice contraction could be analyzed.

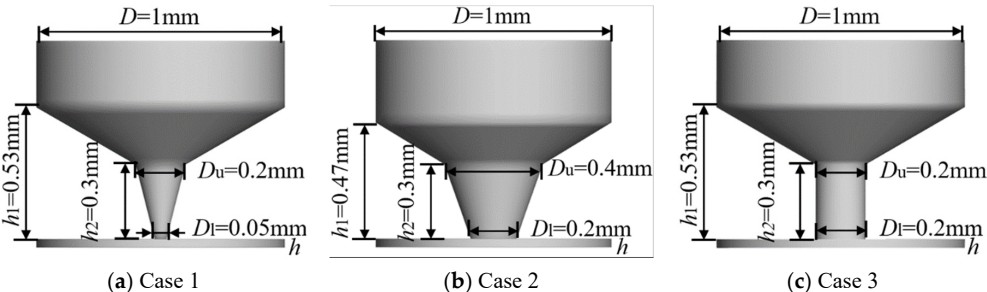

(**a**) Case 1      (**b**) Case 2      (**c**) Case 3

**Figure 4.** Sketches of the geometrical parameters of the aerostatic bearings.

**Table 2.** Geometrical parameters of the aerostatic bearings.

| Parameter | Case 1 | Case 2 | Case 3 |
| --- | --- | --- | --- |
| $D_u$ | 0.2 mm | 0.4 mm | 0.2 mm |
| $D_l$ | 0.05 mm | 0.2 mm | 0.2 mm |
| $h_1$ | 0.53 mm | 0.47 mm | 0.53 mm |
| $h_2$ | 0.3 mm | 0.3 mm | 0.3 mm |

The inlet pressure is 0.5 MPa for all the cases. Since the orifice obviously contracts, the velocity should be much higher at the orifice outlet. Since the Reynolds number of aerostatic bearing is relatively low here, direct numerical simulation (DNS) was conducted without any turbulence model. The structured grid system was constructed in the present study. "O-H" Topology was adopted for the grid generation in line with the geometry of the aerodynamic bearing, as shown in Figure 4b. First, the height of the first layer of the mesh is $2.7 \times 10^{-6}$ m according to the $y^+ = 1$ requirement and the growth ratio $\approx 1.15$ to guarantee the grid resolution within the boundary layer. There are 80 and 90 grids in the circumferential and axial directions, respectively. In detail, there are 30 grids in the normal direction of clearance to resolve the flow structures. Consequently, the overall grid number is around 0.27 M, the side and vertical views of which are illustrated in Figure 5. The non-reflection condition was imposed on the outlets of bearings. The grid generation was accomplished with PointwiseV18.3R1 software.

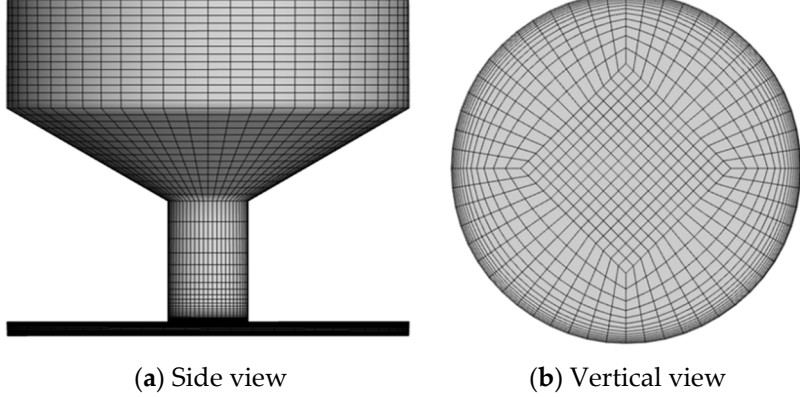

(**a**) Side view      (**b**) Vertical view

**Figure 5.** Sketches of the computational mesh.

## 7. Numerical Results and analysis

As a result, a vertical slice is extracted across the axis line. The distributions of the static pressure and Mach numbers of Case 1 and Case 2 are demonstrated in Figures 6 and 7, respectively. The influence of orifice contraction could be discussed. The flow mechanism is quite delicate as shown in Figure 6b. In detail, the flow becomes supersonic at the outlet of the orifice. Then, the velocity magnitude dramatically decreases near the stagnation point with an increase in static pressure, as shown in Figure 6a. At the entrance of the air film, the flow becomes supersonic again within a short distance due to the geometrical contraction. It turns out that the static pressure in the air film is relatively low, leading to poor bearing capacity.

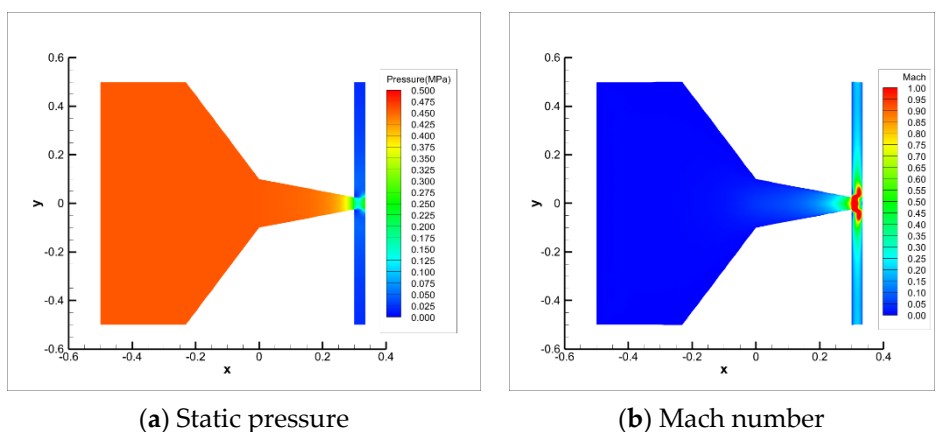

(**a**) Static pressure            (**b**) Mach number

**Figure 6.** Physical quantity distributions on the vertical slice of Case 1.

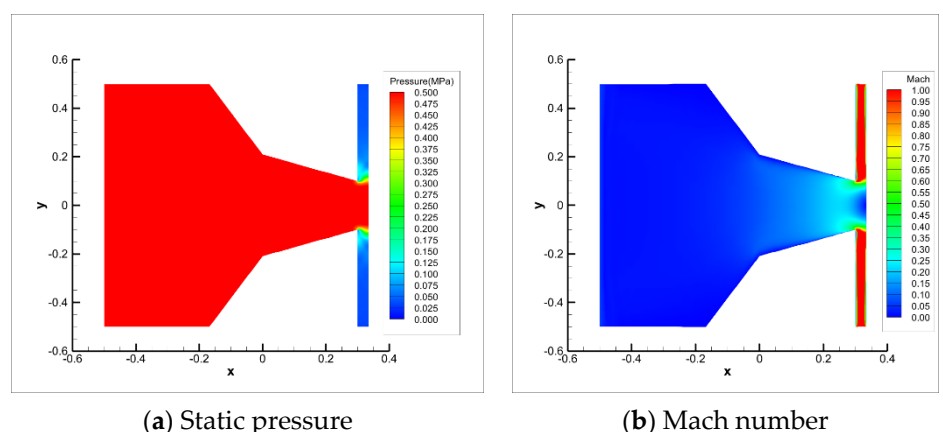

(**a**) Static pressure            (**b**) Mach number

**Figure 7.** Physical quantity distributions on the vertical slice of Case 2.

With a gentle contraction compared with Case 1, the acceleration inside the orifice is not acute in Case 2, as shown in Figure 7b. However, the velocity magnitude resides in the supersonic regime in the air film and the static pressure suddenly decreases at the inlet of the air film, as shown in Figure 7a, indicating a great pressure gradient.

Case 3 could be regarded as an intermediate state between Case 1 and Case 2, with the same $D_{\mathrm{u}}$ of Case 1 and the same $D_{\mathrm{l}}$ of Case 2. As a result, a supersonic region around the film inlet could be identified as shown in Figure 8b, which could be interpreted with the abrupt contraction. High pressure could be found near the stagnation point and an obvious positive pressure gradient emerges inside the film, as shown in Figure 8a. Due to the combined effects of the pressure gradient and viscosity, the variation of velocity magnitude along the air film is complicated. It decreases near the inlet of the film and accelerates to a supersonic regime near the outlet.

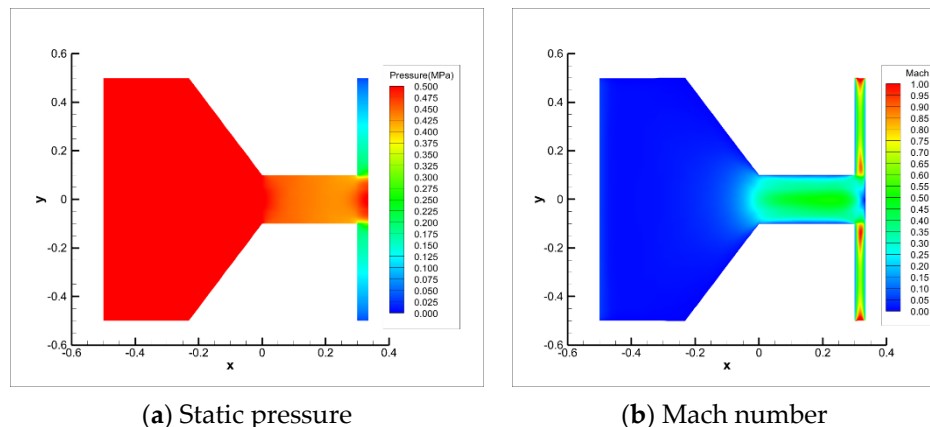

**(a)** Static pressure        **(b)** Mach number

**Figure 8.** Physical quantity distributions on the vertical slice of Case 3.

Pressure distribution curves along the mid-plane of the air film from the symmetrical axis to the outlet, which are extracted from the above figures, are shown in Figure 9. The critical pressure is an important value for analyzing the nozzle flow field, which is the key point from subsonic to supersonic conditions. Since the velocity at the inlet of the air bearing is small, the pressure at the inlet could be regarded as the stagnation pressure. Regardless of the influence of the boundary layer, the air bearing can be approximately regarded as a one-dimensional isentropic flow before the contraction. Therefore, the critical pressure ratio can be analytically given in the form $\frac{p_{cr}}{p^*} = \left(\frac{2}{\gamma+1}\right)^{\frac{\gamma}{\gamma-1}}$, where $p_{cr}$ is the critical pressure and $p^*$ is the stagnation pressure. Since $\gamma = 1.4$ in the present study, the critical pressure ratio is 0.528, which is consistent with our numerical results in Figure 9. Therefore, the numerical accuracy could be validated to some extent. Corresponding to the pressure distribution mentioned above, an obvious pressure gradient emerges near the inlet of the air film at 0.1 R in Case 2 and Case 3. Although the peak value at the stagnation point is higher in Case 2, the overall effect of the static pressure is more pronounced in Case 3 due to the contribution inside the film. The performance of Case 1 is poor due to an early transition to the supersonic regime before the entrance of the air film.

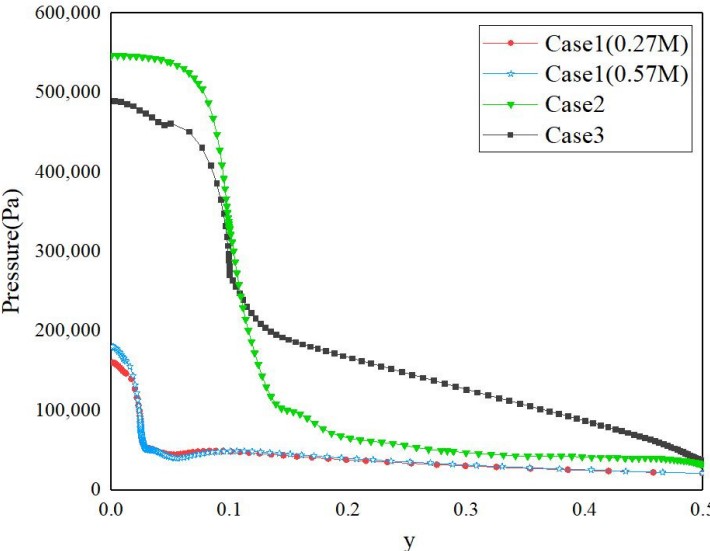

**Figure 9.** Pressure distribution curves in air film.

In addition, an attempt with refined grid resolution has also been delivered. In detail, the number of computational grids is increased to 0.57 M, in contrast with 0.27 M according to the original grid resolution. Actually, the "O-H" topology of the grid system

was maintained, and the grid resolution in all three directions (i.e., the circumferential, axial, and radial directions) has been refined at a ratio of 1.3. The coincidence of the numerical results from the original and refined mesh systems in Figure 9 indicates the grid independence.

As a result, the integration of the pressure upon the thrust surface could be obtained, as shown in Table 3. Consistent with the pressure distribution mentioned above, Case 3 has the best performance in terms of bearing capacity, while the pressure integration of Case 2 is lower. It indicates that orifice contraction has to be accurately optimized, although it could create a higher pressure value near the stagnation point.

**Table 3.** Pressure integration in different cases.

| Parameter | Case 1 | Case 2 | Case 3 |
|:---:|:---:|:---:|:---:|
| Pressure integration(N) | 0.0233 | 0.0813 | 0.0826 |

The local distributions of the Mach number in the vicinity of the air film inlet are shown in Figure 10. Coincident with the above conclusions, the flow structures in Case 1 are more complicated due to the compact geometry and flow interaction. The low-speed region in Case 3 is relatively inconspicuous, indicating a minor loss of kinetic energy.

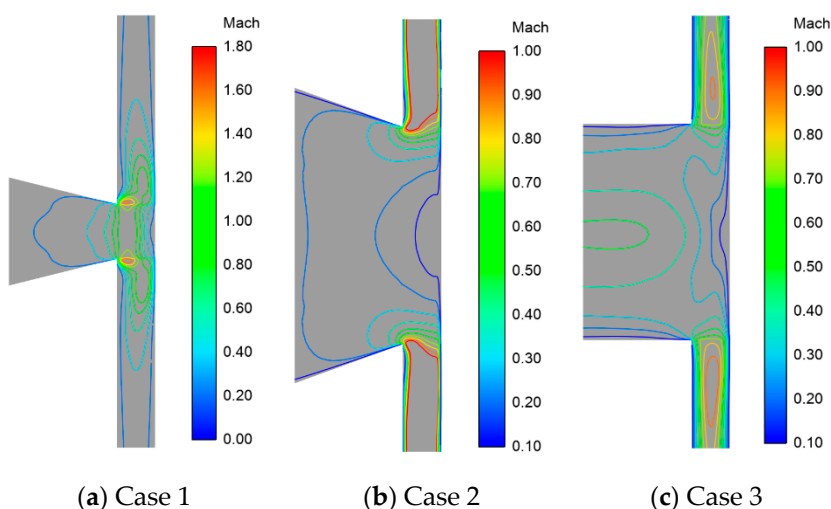

(**a**) Case 1          (**b**) Case 2          (**c**) Case 3

**Figure 10.** Mach number distribution near the air film inlet.

## 8. Conclusions

Since the static and dynamic performances are strongly associated with the flow field of aerostatic bearing, a high-precision numerical method is beneficial for the detailed description of the bearing flow field. Among the candidates, the high-order characteristics and the convergence of DGM have been proven. To this end, a modified IPDG method was introduced here. Meanwhile, a lift operator was included to eliminate the fourth-order homogeneity tensor connecting the viscous term and variable gradient, which could somehow improve the numerical feasibility. The accuracy of the above numerical method has been comprehensively validated through viscous cases, including Couette flow, shear-driven cavity flow, and supersonic airfoil flow field.

Then, the flow fields of three aerostatic bearings were simulated with different orifice geometries. As a result, the distributions of Mach number and static pressure could be estimated together with the integration of the pressure acting upon the thrust surface. Meanwhile, the pressure distribution curve in the air film is also calculated.

The tendency could be extracted to evaluate the influence of orifice geometry. It turns out that the orifice without contraction has the best bearing capacity. The critical point is

that geometrical contraction might lead to an early transition to supersonic flow within the orifice.

Actually, for higher Re, IPDG simulations with LES models would be expected in the near future with the help of experimental references. Therefore, the feasibility and accuracy of the present framework would be thoroughly validated. Meanwhile, the influence of the film thickness would be systematically discussed. Eventually, the flow field is in the mixed compressible/incompressible regime; therefore, the resolution of the low Mach number perturbations and the control of dissipation are important [20,21]. Corresponding research has been accomplished in the FVM framework, which could be extended to the IPDG framework in the near future.

**Author Contributions:** Conceptualization, M.Z. and W.H.; methodology, Q.D.; software, J.C.; validation, S.H. and W.H.; formal analysis, J.C. and Y.H.; investigation, Q.D. and M.Z.; resources, S.H.; data curation, J.C.; writing—original draft preparation, Q.D.; writing—review and editing, M.Z.; visualization, S.H. and K.F.; supervision, M.Z. and K.F.; project administration, M.Z. and Y.H.; funding acquisition, W.H. and M.Z. All authors have read and agreed to the published version of the manuscript.

**Funding:** This study is supported by the National Natural Science Foundation of China [Grant No. 11972250, 91952301 and 12102298], the China Postdoctoral Science Foundation [Grant No. 2021M702443], and the China Automotive Technology & Research Center Co. Ltd. [Grant No. ZX20220002].

**Data Availability Statement:** The data presented in this study are available on request from the corresponding author.

**Conflicts of Interest:** The authors declare no conflict of interest.

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
