# Peer review of "High-Precision Flow Field Simulation of Aerostatic Bearings Based on the Interior Penalty Discontinuous Galerkin Method"

_lubricants, doi:10.3390/lubricants10120360_

Round 1
Reviewer 1 Report
1. Please check Fig.5b. The velocity in the clearance is maximal. However, the pressure gradient in Case 3 is maximal due to Fig.7. This is inconsistent with physical principles.
2. The accuracy of IPDG method is not sufficiently proved in the field of airbearing. As described in Section 4, the shear-driven cavity flow is the laminar incompressible flow in a square cavity. The IPDG method can be validated in the laminar incompressible flow. However, the flow in airbearing is compressible flow. How to prove that the IPDG method also has high accuracy in compressible flow?
3. In the Section 5-6, the analysis results should be compared with different methods(such as FVM) and experimental results to prove the accuracy. Otherwise the discussion and conclusion is meaningless.
Reviewer 2 Report
Dear authors,
paper is interesting.
The main concern is the fact I couldn't see that you analyzed choking effect at the outlet section where the Ma=1 (Ma is Mach number) is achieved. The ratio of the pressure in the nozzle, before the contraction, and the critical pressure value in the section where Ma=1, could be analytically determined.
I hope you will discuss this, as you don't have the experimental verification.
Regards,
Reviewer.
Reviewer 3 Report
The following queries needs to be addressed to enhance the merit of this manuscript for journal publication:
1. The paragrapgh from line 86 need not be included. Instead the focus should have been to summarize the lierature review and state the present research objectives with a clear statement of research gaps.
2. The manuscript seems to be an extended version of conference paper and seems short of a full fledged journal article in depth and content. This is in no way demeaning the article. The article in current form is a good work but not enough for journal article. The authors may extend teh analysis further.
3. The meshing details are shown but the software tool used is not mentioned. Also, grid independence test is missing without which the numerical results are difficult to be accepted.
4. Since, Mach number is seen grater than 1 or atleast equalling 1 in one of the contour plots, does it indicate that the analysis should be compressible flow? is it included in teh analysis?
The work is good but incomplete. I suggest the authors to further the analysis.
Reviewer 4 Report
The authors present numerical simulations of the flows inside the aerostatic bearings. The paper is interesting and could be published subject to the revisions below:
1) The authors should discuss low Mach number effects that affect the simulations Thornber et al. Numerical dissipation of upwind schemes in low Mach flow, International Journal for Numerical Methods in Fluids, Vol. 56, 8, 1535-1541, 2008; An Improved Reconstruction Method for Compressible Flows with Low Mach Number Features, Journal of Computational Physics, 227, 4873-4894, 2008.
2) They should also discuss the uncertainty of the numerical results.
3) Discussion on mesh convergence is also needed.
Reviewer 5 Report
mesh sensitivity study or grid independent test not available. You may add to show how to define the mesh size
Mesh parameter is not specified. Please include
Round 2
Reviewer 1 Report
I reread the whole paper, and the shortcomings of revised paper are shown as follow:
1. The superiority of IPDG method has not been proved. There are several methods for performance analysis of air bearings, such as the early FDM and FEM based on Reynolds equation, the commercial software based on FVM and so on. The advantages of IPDG method applied to the performance analysis of air bearings need to be elaborated. For example, wether the Case2 is a special phenomenon found under the IPDG method, or Fluent and other traditional FVM methods can achieve the same phenomenon. When fluent and other traditional FVM methods can obtain the same phenomenon, what is the significance of using IPDG method.
2. Further evidence is needed to confirm the flow field changes caused by different gas bearing inlet structures. The evidence can be the analysis results based on different numerical methods or the experimental analysis results. The unverified analysis results are of little academic significance.
3. The first item in the conclusion, the verification of the accuracy of the IPDG method is not the main work of this paper and should not be taken as a conclusion.
4. Case study is not enough. At present, only the air flow field under 0.035 mm gas film thickness is analyzed, but the performance of air bearing needs to consider the change of load capacity under different gas film thickness, that is, the stiffness. For example, in case 1, the bearing capacity is the smallest, but its stiffness is the largest in a small film gap, such as 0.01mm. The current analysis results are insufficient to optimize the air inlet structure of gas bearing.
Reviewer 2 Report
Dear Authors,
you have improved the paper and meet some questions, but still the concept is not well.
1. Mesh generation is not adequatly presented.
2. Test case is not discussed in details.
3. Experimental validation is still a weak side.
So, there is still a lot of space for improvement.
Best regards,
Reviewer.
Reviewer 3 Report
All the comments are addressed by authors and looks fine now. I have no further comments.
Author Response
Thank you for your help and approval of the manuscript.
Reviewer 4 Report
The authors have revised the paper, and it can now be published as is.
Author Response

(The authors gave the same response as above.)

Round 3
Reviewer 1 Report
The main suggestions have been answered. Further research results are expected in future work. I agree to accept the paper in present form.
Author Response
Thank you for your help and approval of our manuscript.
Reviewer 2 Report
Dear Editor,
Paper is improved, but you should discuss more on experimental validation in the paper.
Regards,
Reviewer.
Author Response
Dear Reviewer,
Thank you for your kindly suggestions, experimental validation is really important for our study. Now we will accomplish an IPDG simulation together with LES method. Quantitative validation between the numerical and experimental results will be conducted according to your suggestions in the near future.
Thank you for your valuable suggestions.
Yours sincerely,
Ming Zhao